# TinyTL: Reduce Memory, Not Parameters
# for Efficient On-Device Learning

**Han Cai**[1], **Chuang Gan**[2], **Ligeng Zhu**[1], **Song Han**[1]
[1]Massachusetts Institute of Technology,    [2]MIT-IBM Watson AI Lab
http://tinyml.mit.edu/

## Abstract

On-device learning enables edge devices to continually adapt the AI models to new data, which requires a small memory footprint to fit the tight memory constraint of edge devices. Existing work solves this problem by reducing the number of trainable parameters. However, this doesn't directly translate to memory saving since the major bottleneck is the activations, not parameters. In this work, we present *Tiny-Transfer-Learning* (TinyTL) for memory-efficient on-device learning. TinyTL freezes the *weights* while only learns the *bias* modules, thus no need to store the intermediate activations. To maintain the adaptation capacity, we introduce a new memory-efficient bias module, the *lite residual module*, to refine the feature extractor by learning small residual feature maps adding only 3.8% memory overhead. Extensive experiments show that TinyTL significantly saves the memory (up to **6.5×**) with little accuracy loss compared to fine-tuning the full network. Compared to fine-tuning the last layer, TinyTL provides significant accuracy improvements (up to **34.1%**) with little memory overhead. Furthermore, combined with feature extractor adaptation, TinyTL provides **7.3-12.9×** memory saving without sacrificing accuracy compared to fine-tuning the full Inception-V3.

## 1 Introduction

Intelligent edge devices with rich sensors (e.g., billions of mobile phones and IoT devices)[1] have been ubiquitous in our daily lives. These devices keep collecting *new* and *sensitive* data through the sensor every day while being expected to provide high-quality and customized services without sacrificing privacy[2]. These pose new challenges to efficient AI systems that could not only run inference but also continually fine-tune the pre-trained models on newly collected data (i.e., on-device learning).

Though on-device learning can enable many appealing applications, it is an extremely challenging problem. First, edge devices are *memory-constrained*. For example, a Raspberry Pi 1 Model A only has 256MB of memory, which is sufficient for inference, but by far insufficient for training (Figure 1 left), even using a lightweight neural network architecture (MobileNetV2 [1]). Furthermore, the memory is shared by various on-device applications (e.g., other deep learning models) and the operating system. A single application may only be allocated a small fraction of the total memory, which makes this challenge more critical. Second, edge devices are *energy-constrained*. DRAM access consumes two orders of magnitude more energy than on-chip SRAM access. The large memory footprint of activations cannot fit into the limited on-chip SRAM, thus has to access DRAM. For instance, the training memory of MobileNetV2, under batch size 16, is close to 1GB, which is by far larger than the SRAM size of an AMD EPYC CPU[3] (Figure 1 left), not to mention lower-end

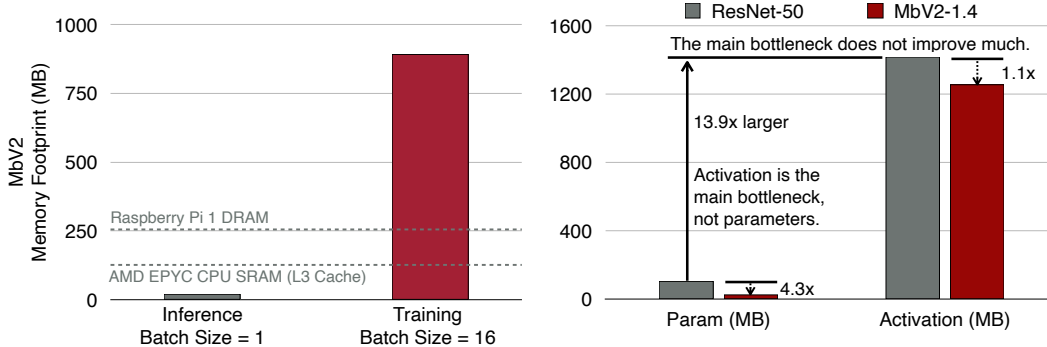

Figure 1: *Left*: The memory footprint required by training is much larger than inference. *Right*: Memory cost comparison between ResNet-50 and MobileNetV2-1.4 under batch size 16. Recent advances in efficient model design only reduce the size of parameters, but the activation size, which is the main bottleneck for training, does not improve much.

edge platforms. If the training memory can fit on-chip SRAM, it will drastically improve the speed and energy efficiency.

There is plenty of efficient inference techniques that reduce the number of trainable parameters and the computation FLOPs [1, 2, 3, 4, 5, 6, 7, 8, 9, 10], however, parameter-efficient or FLOPs-efficient techniques do not directly save the training memory. It is the activation that bottlenecks the training memory, not the parameters. For example, Figure 1 (right) compares ResNet-50 and MobileNetV2-1.4. In terms of parameter size, MobileNetV2-1.4 is $4.3\times$ smaller than ResNet-50. However, for training activation size, MobileNetV2-1.4 is almost the same as ResNet-50 (only $1.1\times$ smaller), leading to little memory reduction. It is essential to reduce the size of intermediate activations required by back-propagation, which is the key memory bottleneck for efficient on-device training.

In this paper, we propose *Tiny-Transfer-Learning* (TinyTL) to address these challenges. By analyzing the memory footprint during the backward pass, we notice that the intermediate activations (the main bottleneck) are only needed when updating the weights, not the biases (Eq. 2). Inspired by this finding, we propose to freeze the weights of the pre-trained feature extractor and only update the biases to reduce the memory footprint (Figure 2b). To compensate for the capacity loss, we introduce a memory-efficient bias module, called *lite residual module*, which improves the model capacity by refining the intermediate feature maps of the feature extractor (Figure 2c). Meanwhile, we aggressively shrink the resolution and width of the lite residual module to have a small memory overhead (only 3.8%). Extensive experiments on 9 image classification datasets with the same pre-trained model (ProxylessNAS-Mobile [11]) demonstrate the effectiveness of TinyTL compared to previous transfer learning methods. Further, combined with a pre-trained once-for-all network [10], TinyTL can select a specialized sub-network as the feature extractor for each transfer dataset (i.e., feature extractor adaptation): given a more difficult dataset, a larger sub-network is selected, and vice versa. TinyTL achieves the same level of (or even higher) accuracy compared to fine-tuning the full Inception-V3 while reducing the training memory footprint by up to **12.9×**. Our contributions can be summarized as follows:

- We propose TinyTL, a novel transfer learning method to reduce the training memory footprint by an order of magnitude for efficient on-device learning. We systematically analyze the memory of training and find the bottleneck comes from updating the weights, not biases (assume ReLU activation).

- We also introduce the *lite residual module*, a memory-efficient bias module to improve the model capacity with little memory overhead.

- Extensive experiments on transfer learning tasks show that our method is highly memory-efficient and effective. It reduces the training memory footprint by up to 12.9× without sacrificing accuracy.

## 2  Related Work

**Efficient Inference Techniques.**  Improving the inference efficiency of deep neural networks on resource-constrained edge devices has recently drawn extensive attention. Starting from [4, 5, 12, 13,

14], one line of research focuses on compressing pre-trained neural networks, including i) network pruning that removes less-important units [4, 15] or channels [16, 17]; ii) network quantization that reduces the bitwidth of parameters [5, 18] or activations [19, 20]. However, these techniques cannot handle the training phase, as they rely on a well-trained model on the target task as the starting point.

Another line of research focuses on lightweight neural architectures by either manual design [1, 2, 3, 21, 22] or neural architecture search [6, 8, 11, 23]. These lightweight neural networks provide highly competitive accuracy [10, 24] while significantly improving inference efficiency. However, concerning the training memory efficiency, key bottlenecks are not solved: **the training memory is dominated by activations, not parameters** (Figure 1).

There are also some non-deep learning methods [25, 26, 27] that are designed for efficient inference on edge devices. These methods are suitable for handling simple tasks like MNIST. However, for more complicated tasks, we still need the representation capacity of deep neural networks.

**Memory Footprint Reduction.**    Researchers have been seeking ways to reduce the training memory footprint. One typical approach is to re-compute discarded activations during backward [28, 29]. This approach reduces memory usage at the cost of a large computation overhead. Thus it is not preferred for edge devices. Layer-wise training [30] can also reduce the memory footprint compared to end-to-end training. However, it cannot achieve the same level of accuracy as end-to-end training. Another representative approach is through activation pruning [31], which builds a dynamic sparse computation graph to prune activations during training. Similarly, [32] proposes to reduce the bitwidth of training activations by introducing new reduced-precision floating-point formats. Besides reducing the training memory cost, there are some techniques that focus on reducing the peak inference memory cost, such as RNNPool [33] and MemNet [34]. Our method is orthogonal to these techniques and can be combined to further reduce the memory footprint.

**Transfer Learning.**    Neural networks pre-trained on large-scale datasets (e.g., ImageNet [35]) are widely used as a fixed feature extractor for transfer learning, then only the last layer needs to be fine-tuned [36, 37, 38, 39]. This approach does not require to store the intermediate activations of the feature extractor, and thus is memory-efficient. However, the capacity of this approach is limited, resulting in poor accuracy, especially on datasets [40, 41] whose distribution is far from ImageNet (e.g., only 45.9% Aircraft top1 accuracy achieved by Inception-V3 [42]). Alternatively, fine-tuning the full network can achieve better accuracy [43, 44]. But it requires a vast memory footprint and hence is not friendly for training on edge devices. Recently, [45, 46] propose to only update parameters of the batch normalization (BN) [47] layers, which greatly reduces the number of trainable parameters. **Unfortunately, parameter-efficiency doesn't translate to memory-efficiency.** It still requires a large amount of memory (e.g., 326MB under batch size 8) to store the input activations of the BN layers (Table 3). Additionally, the accuracy of this approach is still much worse than fine-tuning the full network (70.7% v.s. 85.5%; Table 3). People can also partially fine-tune some layers, but how many layers to select is still ad hoc. This paper provides a systematic approach to save memory without losing accuracy.

## 3  Tiny Transfer Learning

### 3.1  Understanding the Memory Footprint of Back-propagation

Without loss of generality, we consider a neural network $\mathcal{M}$ that consists of a sequence of layers:

$$\mathcal{M}(\cdot) = \mathcal{F}_{\mathbf{w}_n}(\mathcal{F}_{\mathbf{w}_{n-1}}(\cdots \mathcal{F}_{\mathbf{w}_2}(\mathcal{F}_{\mathbf{w}_1}(\cdot))\cdots)), \qquad (1)$$

where $\mathbf{w}_i$ denotes the parameters of the $i^{th}$ layer. Let $\mathbf{a}_i$ and $\mathbf{a}_{i+1}$ be the input and output activations of the $i^{th}$ layer, respectively, and $\mathcal{L}$ be the loss. In the backward pass, given $\frac{\partial \mathcal{L}}{\partial \mathbf{a}_{i+1}}$, there are two goals for the $i^{th}$ layer: computing $\frac{\partial \mathcal{L}}{\partial \mathbf{a}_i}$ and $\frac{\partial \mathcal{L}}{\partial \mathbf{w}_i}$.

Assuming the $i^{th}$ layer is a linear layer whose forward process is given as: $\mathbf{a}_{i+1} = \mathbf{a}_i \mathbf{W} + \mathbf{b}$, then its backward process under batch size 1 is

$$\frac{\partial \mathcal{L}}{\partial \mathbf{a}_i} = \frac{\partial \mathcal{L}}{\partial \mathbf{a}_{i+1}} \frac{\partial \mathbf{a}_{i+1}}{\partial \mathbf{a}_i} = \frac{\partial \mathcal{L}}{\partial \mathbf{a}_{i+1}} \mathbf{W}^T, \qquad \frac{\partial \mathcal{L}}{\partial \mathbf{W}} = \mathbf{a}_i^T \frac{\partial \mathcal{L}}{\partial \mathbf{a}_{i+1}}, \qquad \frac{\partial \mathcal{L}}{\partial \mathbf{b}} = \frac{\partial \mathcal{L}}{\partial \mathbf{a}_{i+1}}. \qquad (2)$$

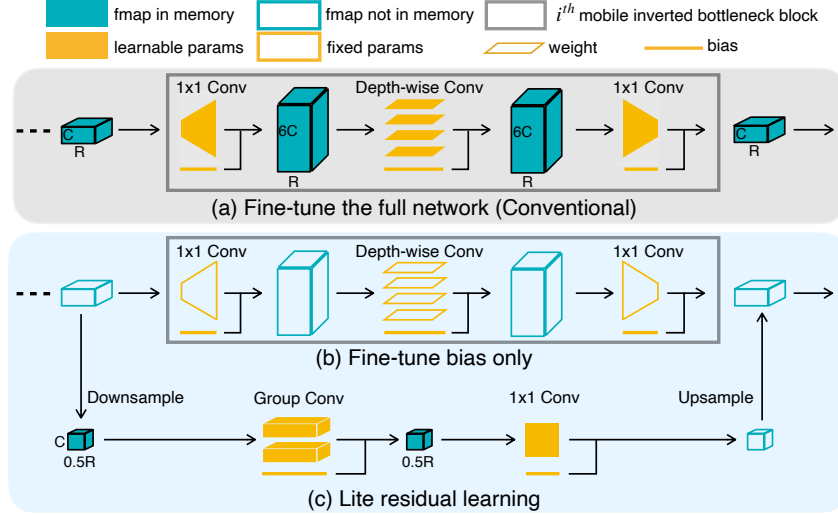

Figure 2: TinyTL overview ("C" denotes the width and "R" denote the resolution). Conventional transfer learning relies on fine-tuning the weights to adapt the model (Fig.a), which requires a large amount of activation memory (in blue) for back-propagation. TinyTL reduces the memory usage by fixing the weights (Fig.b) while only fine-tuning the bias. (Fig.c) exploit *lite residual learning* to compensate for the capacity loss, using group convolution and avoiding inverted bottleneck to achieve high arithmetic intensity and small memory footprint. The skip connection remains unchanged (omitted for simplicity).

According to Eq. (2), the intermediate activations (i.e., $\{\mathbf{a}_i\}$) that dominate the memory footprint are only required to compute the gradient of the weights (i.e., $\frac{\partial \mathcal{L}}{\partial \mathbf{W}}$), not the bias. If we only update the bias, training memory can be greatly saved. This property is also applicable to convolution layers and normalization layers (e.g., batch normalization [47], group normalization [48], etc) since they can be considered as special types of linear layers.

Regarding non-linear activation layers (e.g., ReLU, sigmoid, h-swish), sigmoid and h-swish require to store $\mathbf{a}_i$ to compute $\frac{\partial \mathcal{L}}{\partial \mathbf{a}_i}$ (Table 1), hence they are not memory-efficient. Activation layers that build upon them are also not memory-efficient consequently, such as tanh, swish [49], etc. In contrast, ReLU and other ReLU-styled activation layers (e.g., LeakyReLU [50]) only requires to store a binary mask representing whether the value is smaller than 0, which is $32\times$ smaller than storing $\mathbf{a}_i$.

Table 1: Detailed forward and backward processes of non-linear activation layers. $|\mathbf{a}_i|$ denotes the number of elements of $\mathbf{a}_i$. "$\circ$" denotes the element-wise product. $(\mathbf{1}_{\mathbf{a}_i\geq 0})_j = 0$ if $(\mathbf{a}_i)_j < 0$ and $(\mathbf{1}_{\mathbf{a}_i\geq 0})_j = 1$ otherwise. $\text{ReLU6}(\mathbf{a}_i) = \min(6, \max(0, \mathbf{a}_i))$.

| Layer Type | Forward | Backward | Memory Cost |
|---|---|---|---|
| ReLU | $\mathbf{a}_{i+1} = \max(0, \mathbf{a}_i)$ | $\frac{\partial \mathcal{L}}{\partial \mathbf{a}_i} = \frac{\partial \mathcal{L}}{\partial \mathbf{a}_{i+1}} \circ \mathbf{1}_{\mathbf{a}_i \geq 0}$ | $|\mathbf{a}_i|$ bits |
| sigmoid | $\mathbf{a}_{i+1} = \sigma(\mathbf{a}_i) = \frac{1}{1+\exp(-\mathbf{a}_i)}$ | $\frac{\partial \mathcal{L}}{\partial \mathbf{a}_i} = \frac{\partial \mathcal{L}}{\partial \mathbf{a}_{i+1}} \circ \sigma(\mathbf{a}_i) \circ (1 - \sigma(\mathbf{a}_i))$ | $32\,|\mathbf{a}_i|$ bits |
| h-swish [7] | $\mathbf{a}_{i+1} = \mathbf{a}_i \circ \frac{\text{ReLU6}(\mathbf{a}_i+3)}{6}$ | $\frac{\partial \mathcal{L}}{\partial \mathbf{a}_i} = \frac{\partial \mathcal{L}}{\partial \mathbf{a}_{i+1}} \circ \left( \frac{\text{ReLU6}(\mathbf{a}_i+3)}{6} + \mathbf{a}_i \circ \frac{\mathbf{1}_{-3 \leq \mathbf{a}_i \leq 3}}{6} \right)$ | $32\,|\mathbf{a}_i|$ bits |

## 3.2 Lite Residual Learning

Based on the memory footprint analysis, one possible solution of reducing the memory cost is to freeze the weights of the pre-trained feature extractor while only update the biases (Figure 2b). However, only updating biases has limited adaptation capacity. Therefore, we introduce lite residual learning that exploits a new class of generalized memory-efficient bias modules to refine the intermediate feature maps (Figure 2c).

Formally, a layer with frozen weights and learnable biases can be represented as:

$$\mathbf{a}_{i+1} = \mathcal{F}_{\mathbf{W}}(\mathbf{a}_i) + \mathbf{b}. \tag{3}$$

To improve the model capacity while keeping a small memory footprint, we propose to add a lite residual module that generates a residual feature map to refine the output:

$$\mathbf{a}_{i+1} = \mathcal{F}_{\mathbf{W}}(\mathbf{a}_i) + \mathbf{b} + \mathcal{F}_{\mathbf{w}_r}(\mathbf{a}_i' = \text{reduce}(\mathbf{a}_i)), \tag{4}$$

where $\mathbf{a}_i' = \text{reduce}(\mathbf{a}_i)$ is the reduced activation. According to Eq. (2), learning these lite residual modules only requires to store the reduced activations $\{\mathbf{a}_i'\}$ rather than the full activations $\{\mathbf{a}_i\}$.

**Implementation (Figure 2c).**   We apply Eq. (4) to mobile inverted bottleneck blocks (MB-block) [1]. The key principle is to keep the activation small. Following this principle, we explore two design dimensions to reduce the activation size:

- **Width.** The widely-used inverted bottleneck requires a huge number of channels ($6\times$) to compensate for the small capacity of a depthwise convolution, which is parameter-efficient but highly activation-inefficient. Even worse, converting $1\times$ channels to $6\times$ channels back and forth requires two $1 \times 1$ projection layers, which doubles the total activation to $12\times$. Depthwise convolution also has a very low arithmetic intensity (its OPs/Byte is less than 4% of $1 \times 1$ convolution's OPs/Byte if with 256 channels), thus highly memory in-efficient with little reuse. To solve these limitations, our lite residual module employs the group convolution that has much higher arithmetic intensity than depthwise convolution, providing a good trade-off between FLOPs and memory. That also removes the $1\times1$ projection layer, reducing the total channel number by $\frac{6\times2+1}{1+1} = 6.5\times$.

- **Resolution.** The activation size grows quadratically with the resolution. Therefore, we shrink the resolution in the lite residual module by employing a $2 \times 2$ average pooling to downsample the input feature map. The output of the lite residual module is then upsampled to match the size of the main branch's output feature map via bilinear upsampling. Combining resolution and width optimizations, the activation of our lite residual module is roughly $2^2 \times 6.5 = 26\times$ smaller than the inverted bottleneck.

## 3.3   Discussions

**Normalization Layers.**   As discussed in Section 3.1, TinyTL flexibly supports different normalization layers, including batch normalization (BN), group normalization (GN), layer normalization (LN), and so on. In particular, BN is the most widely used one in vision tasks. However, BN requires a large batch size to have accurate running statistics estimation during training, which is not suitable for on-device learning where we want a small training batch size to reduce the memory footprint. Moreover, the data may come in a streaming fashion in on-device learning, which requires a training batch size of 1. In contrast to BN, GN can handle a small training batch size as the running statistics in GN are computed independently for different inputs. In our experiments, GN with a small training batch size (e.g., 8) performs slightly worse than BN with a large training batch size (e.g., 256). However, as we target at on-device learning, we choose GN in our models.

**Feature Extractor Adaptation.**   TinyTL can be applied to different backbone neural networks, such as MobileNetV2 [1], ProxylessNASNets [11], EfficientNets [24], etc. However, since the weights of the feature extractor are frozen in TinyTL, we find using the same backbone neural network for all transfer tasks is sub-optimal. Therefore, we choose the backbone of TinyTL using a pre-trained once-for-all network [10] to adaptively select the specialized feature extractor that best fits the target transfer dataset. Specifically, a once-for-all network is a special kind of neural network that is sparsely activated, from which many different sub-networks can be derived without retraining by sparsely activating parts of the model according to the architecture configuration (i.e., depth, width, kernel size, resolution), while the weights are shared. This allows us to efficiently evaluate the effectiveness of a backbone neural network on the target transfer dataset without the expensive pre-training process. Further details of the feature extractor adaptation process are provided in Appendix A.

Table 2: Comparison between TinyTL and conventional transfer learning methods (training memory footprint is calculated assuming the batch size is 8 and the classifier head for Flowers is used). For object classification datasets, we report the top1 accuracy (%) while for CelebA we report the average top1 accuracy (%) over 40 facial attribute classification tasks. 'B' represents Bias while 'L' represents LiteResidual. *FT-Last* represents only the last layer is fine-tuned. *FT-Norm+Last* represents normalization layers and the last layer are fine-tuned. *FT-Full* represents the full network is fine-tuned. The backbone neural network is ProxylessNAS-Mobile, and the resolution is 224 except for 'TinyTL-L+B@320' whose resolution is 320. TinyTL consistently outperforms *FT-Last* and *FT-Norm+Last* by a large margin with a similar or lower training memory footprint. By increasing the resolution to 320, TinyTL can reach the same level of accuracy as *FT-Full* while being 6× memory efficient.

| Method | Train. Mem. | Flowers | Cars | CUB | Food | Pets | Aircraft | CIFAR10 | CIFAR100 | CelebA |
|---|---|---|---|---|---|---|---|---|---|---|
| FT-Last | **31MB** | 90.1 | 50.9 | 73.3 | 68.7 | 91.3 | 44.9 | 85.9 | 68.8 | 88.7 |
| TinyTL-B | **32MB** | 93.5 | 73.4 | 75.3 | 75.5 | 92.1 | 63.2 | 93.7 | 78.8 | 90.4 |
| TinyTL-L | **37MB** | 95.3 | 84.2 | 76.8 | 79.2 | 91.7 | 76.4 | 96.1 | 80.9 | 91.2 |
| TinyTL-L+B | **37MB** | 95.5 | 85.0 | 77.1 | 79.7 | 91.8 | 75.4 | 95.9 | 81.4 | 91.2 |
| TinyTL-L+B@320 | 65MB | 96.8 | 88.8 | 81.0 | 82.9 | 92.9 | 82.3 | 96.1 | 81.5 | - |
| FT-Norm+Last | 192MB | 94.3 | 77.9 | 76.3 | 77.0 | 92.2 | 68.1 | 94.8 | 80.2 | 90.4 |
| FT-Full | 391MB | 96.8 | 90.2 | 81.0 | 84.6 | 93.0 | 86.0 | 97.1 | 84.1 | 91.4 |

## 4 Experiments

### 4.1 Setups

**Datasets.** Following the common practice [43, 44, 45], we use ImageNet [35] as the pre-training dataset, and then transfer the models to 8 downstream object classification tasks, including Cars [41], Flowers [51], Aircraft [40], CUB [52], Pets [53], Food [54], CIFAR10 [55], and CIFAR100 [55]. Besides object classification, we also evaluate our TinyTL on human facial attribute classification tasks, where CelebA [56] is the transfer dataset and VGGFace2 [57] is the pre-training dataset.

**Model Architecture.** To justify the effectiveness of TinyTL, we first apply TinyTL and previous transfer learning methods to the same backbone neural network, ProxylessNAS-Mobile [11]. For each MB-block in ProxylessNAS-Mobile, we insert a lite residual module as described in Section 3.2 and Figure 2 (c). The group number is 2, and the kernel size is 5. We use the ReLU activation since it is more memory-efficient according to Section 3.1. We replace all BN layers with GN layers to better support small training batch sizes. We set the number of channels per group to 8 for all GN layers. Following [58], we apply weight standardization [59] to convolution layers that are followed by GN.

For feature extractor adaptation, we build the once-for-all network using the MobileNetV2 design space [10, 11] that contains five stages with a gradually decreased resolution, and each stage consists of a sequence of MB-blocks. In the stage-level, it supports elastic depth (i.e., 2, 3, 4). In the block-level, it supports elastic kernel size (i.e., 3, 5, 7) and elastic width expansion ratio (i.e., 3, 4, 6). Similarly, for each MB-block in the once-for-all network, we insert a lite residual module that supports elastic group number (i.e., 2, 4) and elastic kernel size (i.e., 3, 5).

**Training Details.** We freeze the memory-heavy modules (weights of the feature extractor) and only update memory-efficient modules (bias, lite residual, classifier head) during transfer learning. The models are fine-tuned for 50 epochs using the Adam optimizer [60] with batch size 8 on a single GPU. The initial learning rate is tuned for each dataset while cosine schedule [61] is adopted for learning rate decay. We apply 8bits weight quantization [5] on the frozen weights to reduce the parameter size, which causes a negligible accuracy drop in our experiments. For all compared methods, we also assume the 8bits weight quantization is applied if eligible when calculating their training memory footprint. Additionally, as PyTorch does not support explicit fine-grained memory management, we use the theoretically calculated training memory footprint for comparison in our experiments. For simplicity, we assume the batch size is 8 for all compared methods throughout the experiment section.

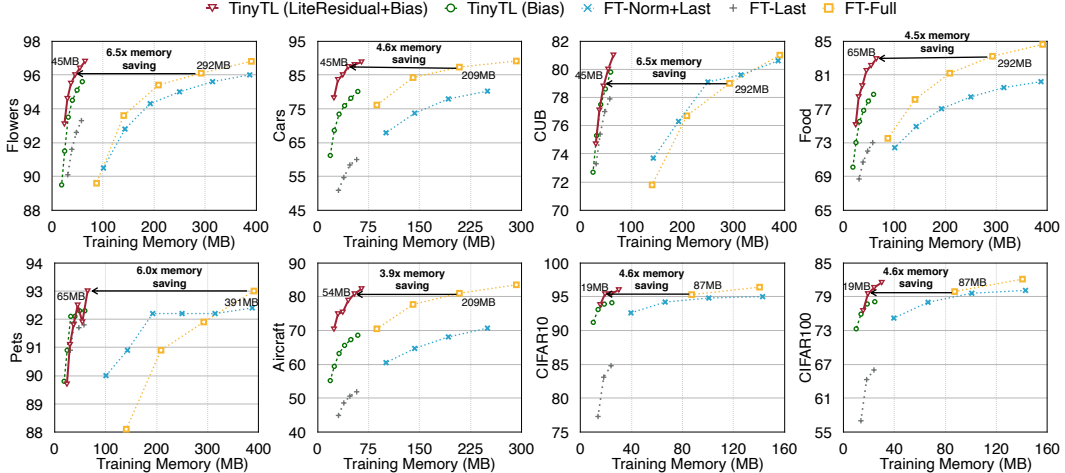

Figure 3: Top1 accuracy results of different transfer learning methods under varied resolutions using the same pre-trained neural network (ProxylessNAS-Mobile). With the same level of accuracy, TinyTL achieves 3.9-6.5× memory saving compared to fine-tuning the full network.

## 4.2 Main Results

**Effectiveness of TinyTL.** Table 2 reports the comparison between TinyTL and previous transfer learning methods including: i) fine-tuning the last linear layer [36, 37, 39] (referred to as *FT-Last*); ii) fine-tuning the normalization layers (e.g., BN, GN) and the last linear layer [42] (referred to as *FT-Norm+Last*) ; iii) fine-tuning the full network [43, 44] (referred to as *FT-Full*). We also study several variants of TinyTL including: i) TinyTL-B that fine-tunes biases and the last linear layer; ii) TinyTL-L that fine-tunes lite residual modules and the last linear layer; iii) TinyTL-L+B that fine-tunes lite residual modules, biases, and the last linear layer. All compared methods use the same pre-trained model but fine-tune different parts of the model as discussed above. We report the average accuracy across five runs.

Compared to *FT-Last*, TinyTL maintains a similar training memory footprint while improving the top1 accuracy by a significant margin. In particular, TinyTL-L+B improves the top1 accuracy by **34.1%** on Cars, by **30.5%** on Aircraft, by **12.6%** on CIFAR100, by **11.0%** on Food, etc. It shows the improved adaptation capacity of our method over *FT-Last*. Compared to *FT-Norm+Last*, TinyTL-L+B improves the training memory efficiency by **5.2×** while providing up to **7.3%** higher top1 accuracy, which shows that our method is not only more memory-efficient but also more effective than *FT-Norm+Last*. Compared to *FT-Full*, TinyTL-L+B@320 can achieve the same level of accuracy while providing **6.0×** training memory saving.

Regarding the comparison between different variants of TinyTL, both TinyTL-L and TinyTL-L+B have clearly better accuracy than TinyTL-B while incurring little memory overhead. It shows that the lite residual modules are essential in TinyTL. Besides, we find that TinyTL-L+B is slightly better than TinyTL-L on most of the datasets while maintaining the same memory footprint. Therefore, we choose TinyTL-L+B as the default.

Figure 3 demonstrates the results under different input resolutions. We can observe that simply reducing the input resolution will result in significant accuracy drops for *FT-Full*. In contrast, TinyTL can reduce the memory footprint by **3.9-6.5×** while having the same or even higher accuracy compared to fine-tuning the full network.

**Combining TinyTL and Feature Extractor Adaptation.** Table 3 summarizes the results of TinyTL and previously reported transfer learning results, where different backbone neural networks are used as the feature extractor. Combined with feature extractor adaptation, TinyTL achieves **7.5-12.9×** memory saving compared to fine-tuning the full Inception-V3, reducing from 850MB to 66-114MB while providing the same level of accuracy. Additionally, we try updating the last two layers besides biases and lite residual modules (indicated by [†]), which results in 2MB of extra

Table 3: Comparison with previous transfer learning results under different backbone neural networks. 'I-V3' is Inception-V3; 'N-A' is NASNet-A Mobile; 'M2-1.4' is MobileNetV2-1.4; 'R-50' is ResNet-50; 'PM' is ProxylessNAS-Mobile; 'FA' represents feature extractor adaptation. † indicates the last two layers are updated besides biases and lite residual modules in TinyTL. TinyTL+FA reduces the training memory by **7.5-12.9×** without sacrificing accuracy compared to fine-tuning the widely used Inception-V3.

| Method | Net | Train. mem. | Reduce ratio | Flowers | Cars | CUB | Food | Pets | Aircraft | CIFAR10 | CIFAR100 |
|---|---|---|---|---|---|---|---|---|---|---|---|
| FT-Full | I-V3 [44] | 850MB | 1.0× | 96.3 | 91.3 | 82.8 | 88.7 | - | 85.5 | - | - |
| | R-50 [43] | 802MB | 1.1× | 97.5 | 91.7 | - | 87.8 | 92.5 | 86.6 | 96.8 | 84.5 |
| | M2-1.4 [43] | 644MB | 1.3× | 97.5 | 91.8 | - | 87.7 | 91.0 | 86.8 | 96.1 | 82.5 |
| | N-A [43] | 566MB | 1.5× | 96.8 | 88.5 | - | 85.5 | 89.4 | 72.8 | 96.8 | 83.9 |
| FT-Norm+Last | I-V3 [42] | 326MB | 2.6× | 90.4 | 81.0 | - | - | - | 70.7 | - | - |
| FT-Last | I-V3 [42] | 94MB | 9.0× | 84.5 | 55.0 | - | - | - | 45.9 | - | - |
| TinyTL | PM@320 | **65MB** | **13.1×** | 96.8 | 88.8 | 81.0 | 82.9 | 92.9 | 82.3 | 96.1 | 81.5 |
| | FA@256 | **66MB** | 12.9× | 96.8 | 89.6 | 80.8 | 83.4 | 93.0 | 82.4 | 96.8 | 82.7 |
| | FA@352 | 114MB | 7.5× | 97.4 | 90.7 | 82.4 | 85.0 | 93.4 | 84.8 | - | - |
| | FA@352† | 116MB | 7.3× | - | 91.5 | - | 86.0 | - | 85.4 | - | - |

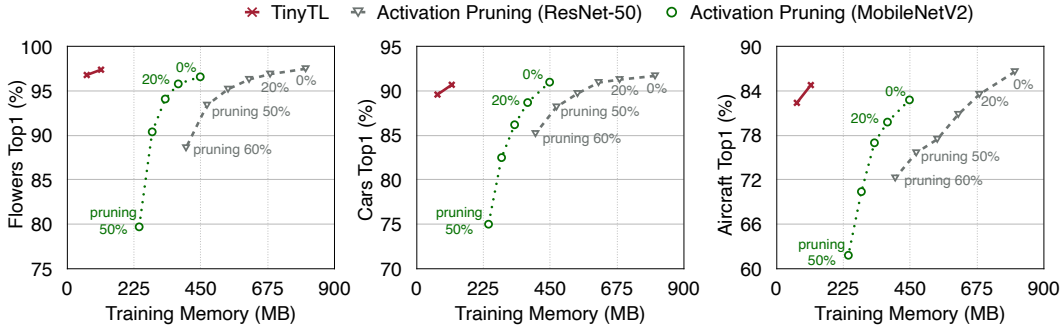

Figure 4: Compared with the dynamic activation pruning [31], TinyTL saves the memory more effectively.

training memory footprint. This slightly improves the accuracy performances, from 90.7% to 91.5% on Cars, from 85.0% to 86.0% on Food, and from 84.8% to 85.4% on Aircraft.

## 4.3 Ablation Studies and Discussions

**Comparison with Dynamic Activation Pruning.** The comparison between TinyTL and dynamic activation pruning [31] is summarized in Figure 4. TinyTL is more effective because it re-designed the transfer learning framework (lite residual module, feature extractor adaptation) rather than prune an existing architecture. The transfer accuracy drops quickly when the pruning ratio increases beyond 50% (only 2× memory saving). In contrast, TinyTL can achieve much higher memory reduction without loss of accuracy.

**Initialization for Lite Residual Modules.** By default, we use the pre-trained weights on the pre-training dataset to initialize the lite residual modules. It requires to have lite residual modules during both the pre-training phase and transfer learning phase. When applying TinyTL to existing pre-trained neural networks that do not have lite residual modules during the pre-training phase, we need to use another initialization strategy for the lite residual modules during transfer learning. To verify the effectiveness of TinyTL under this setting, we also evaluate the performances of TinyTL when using random weights [62] to initialize the lite residual modules except for the scaling parameter of the final normalization layer in each lite residual module. These scaling parameters are initialized with zeros.

Table 4 reports the summarized results. We find using the pre-trained weights to initialize the lite residual modules consistently outperforms using random weights. Besides, we also find that using TinyTL-RandomL+B still provides highly competitive results on Cars, Food, Aircraft, CIFAR10,

Table 4: Results of TinyTL under different initialization strategies for lite residual modules. TinyTL-L+B adds lite residual modules starting from the pre-training phase and uses the pre-trained weights to initialize the lite residual modules during transfer learning. In contrast, TinyTL-RandomL+B uses random weights to initialize the lite residual modules. Using random weights for initialization hurts the performances of TinyTL. But on datasets whose distribution is far from the pre-training dataset, TinyTL-RandomL+B still provides competitive results.

| Method | Train. Mem. | Flowers | Cars | CUB | Food | Pets | Aircraft | CIFAR10 | CIFAR100 | CelebA |
|---|---|---|---|---|---|---|---|---|---|---|
| FT-Last | **31MB** | 90.1 | 50.9 | 73.3 | 68.7 | 91.3 | 44.9 | 85.9 | 68.8 | 88.7 |
| TinyTL-RandomL+B | **37MB** | 88.0 | 82.4 | 72.9 | 79.3 | 84.3 | 73.6 | 95.7 | 81.4 | 91.2 |
| TinyTL-L+B | **37MB** | 95.5 | 85.0 | 77.1 | 79.7 | 91.8 | 75.4 | 95.9 | 81.4 | 91.2 |
| FT-Norm+Last | 192MB | 94.3 | 77.9 | 76.3 | 77.0 | 92.2 | 68.1 | 94.8 | 80.2 | 90.4 |
| FT-Full | 391MB | 96.8 | 90.2 | 81.0 | 84.6 | 93.0 | 86.0 | 97.1 | 84.1 | 91.4 |

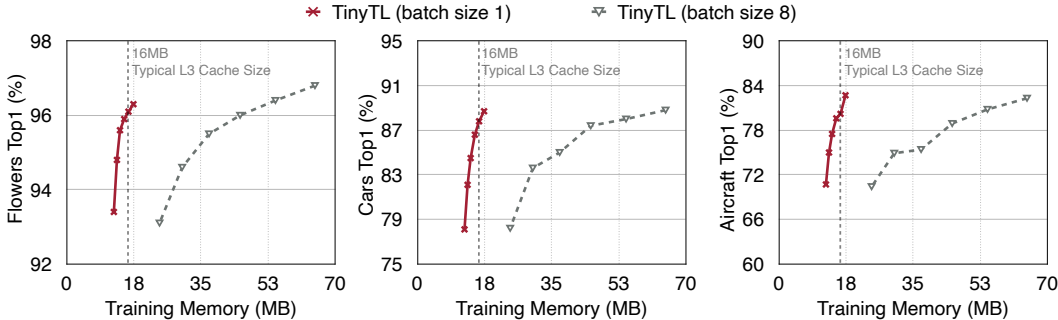

Figure 5: Results of TinyTL when trained with batch size 1. It further reduces the training memory footprint to around 16MB (typical L3 cache size), making it possible to train on the cache (SRAM) instead of DRAM.

CIFAR100, and CelebA. Therefore, if having the budget, it is better to use pre-trained weights to initialize the lite residual modules. If not, TinyTL can still be applied and provides competitive results on datasets whose distribution is far from the pre-training dataset.

**Results of TinyTL under Batch Size 1.** Figure 5 demonstrates the results of TinyTL when using a training batch size of 1. We tune the initial learning rate for each dataset while keeping the other training settings unchanged. As our model employs group normalization rather than batch normalization (Section 3.3), we observe little/no loss of accuracy than training with batch size 8. Meanwhile, the training memory footprint is further reduced to around 16MB, a typical L3 cache size. This makes it much easier to train on the cache (SRAM), which can greatly reduce energy consumption than DRAM training.

## 5    Conclusion

We proposed Tiny-Transfer-Learning (TinyTL) for memory-efficient on-device learning that aims to adapt pre-trained models to newly collected data on edge devices. Unlike previous methods that focus on reducing the number of parameters or FLOPs, TinyTL directly optimizes the training memory footprint by fixing the memory-heavy modules (i.e., weights) while learning memory-efficient bias modules. We further introduce lite residual modules that significantly improve the adaptation capacity of the model with little memory overhead. Extensive experiments on benchmark datasets consistently show the effectiveness and memory-efficiency of TinyTL, paving the way for efficient on-device machine learning.

## Broader Impact

The proposed efficient on-device learning technique greatly reduces the training memory footprint of deep neural networks, enabling adapting pre-trained models to new data locally on edge devices without leaking them to the cloud. It can democratize AI to people in the rural areas where the Internet is unavailable or the network condition is poor. They can not only inference but also fine-tune AI models on their local devices without connections to the cloud servers. This can also benefit privacy-sensitive AI applications, such as health care, smart home, and so on.

## Acknowledgements

We thank MIT-IBM Watson AI Lab, NSF CAREER Award #1943349 and NSF Award #2028888 for supporting this research. We thank MIT Satori cluster for providing the computation resource.

## Footnotes

[1]https://www.statista.com/statistics/330695/number-of-smartphone-users-worldwide/

[2]https://ec.europa.eu/info/law/law-topic/data-protection_en

[3]https://www.amd.com/en/products/cpu/amd-epyc-7302

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
