[Supplementary Material]

# A  Details of Feature Extractor Adaptation

Conventional transfer learning chooses the feature extractor according to the pre-training accuracy (e.g., ImageNet accuracy) and uses the same one for all transfer tasks [1, 2]. However, we find this approach sub-optimal since different target tasks may need very different feature extractors, and high pre-training accuracy does not guarantee good transferability of the pre-trained weights. This is especially critical in our case where the weights are frozen.

Figure 1: Transfer learning performances of various ImageNet pre-trained models with the last linear layer trained. The relative accuracy order between different pre-trained models changes significantly among ImageNet and the transfer learning datasets.

Figure 1 shows the top1 accuracy of various widely used ImageNet pre-trained models on three transfer datasets by only learning the last layer, which reflects the transferability of their pre-trained weights. The relative order between different pre-trained models is not consistent with their ImageNet accuracy on all three datasets. This result indicates that the ImageNet accuracy is not a good proxy for transferability. Besides, we also find that the same pre-trained model can have very different rankings on different tasks. For instance, Inception-V3 gives poor accuracy on Flowers but provides top results on the other two datasets.

Therefore, we need to specialize the feature extractor to best match the target dataset. In this work, we achieve this by using a pre-trained once-for-all network [3] that comprises many different sub-networks. Specifically, given a pre-trained once-for-all network on ImageNet, we fine-tune it on the target transfer dataset with the weights of the main branches (i.e., MB-blocks) frozen and the other parameters (i.e., biases, lite residual modules, classifier head) updated via gradient descent. In this phase, we randomly sample one sub-network in each training step. The peak memory cost of this phase is 61MB under resolution 224, which is reached when the largest sub-network is sampled. Regarding the computation cost, the average MAC (forward & backward)[1] of sampled sub-nets is (776M + 2510M) / 2 = 1643M per sample, where 776M is the training MAC of the smallest sub-network and 2510M is the training MAC of the largest sub-network. Therefore, the total MAC of this phase is 1643M × 2040 × 0.8 × 50 = 134T on Flowers, where 2040 is the number of total training samples, 0.8 means the once-for-all network is fine-tuned on 80% of the training samples (the remaining 20% is reserved for search), and 50 is the number of training epochs.

Based on the fine-tuned once-for-all network, we collect 500 [sub-net, accuracy] pairs on the validation set (20% randomly sampled training data) and train an accuracy predictor[2] using the collected data [3]. We employ evolutionary search [4] based on the accuracy predictor to find the sub-network that best matches the target transfer dataset. No back-propagation on the once-for-all network is required in this step, thus incurs no additional memory overhead. The primary computation cost of this phase comes from collecting 500 [sub-net, accuracy] pairs required to train the accuracy predictor. It only involves the forward processes of sampled sub-nets, and no back-propagation is required. The average MAC (only forward) of sampled sub-nets is (355M + 1182M) / 2 = 768.5M per sample, where 355M is the inference MAC of the smallest sub-network and 1182M is the inference MAC of the largest sub-network. Therefore, the total MAC of this phase is 768.5M × 2040 × 0.2 × 500 = 157T on Flowers, where 2040 is the number of total training samples, 0.2 means the validation set consists of 20% of the training samples, and 500 is the number of measured sub-nets.

Finally, we fine-tune the searched sub-network with the weights of the main branches frozen and the other parameters updated, using the full training set to get the final results. The memory cost of this

Figure 2: On-device training cost on Pets. TinyTL requires $9.9\times$ smaller memory cost (assuming using the same batch size) and $27\times$ smaller computation cost compared to fine-tuning the full MobileNetV2-1.4 [5] while having a better accuracy.

phase is 66MB under resolution 256 on Flowers. The total MAC is 2190M $\times$ 2040 $\times$ 1.0 $\times$ 50 = 223T, on Flowers, where 2190M is the training MAC, 2040 is the number of total training samples, 1.0 means the full training set is used, and 50 is the number of training epochs.

# B   Details of the Accuracy Predictor

The accuracy predictor is a three-layer feed-forward neural network with a hidden dimension of 400 and ReLU as the activation function for each layer. It takes the one-hot encoding of the sub-network's architecture as the input and outputs the predicted accuracy of the given sub-network. The inference MAC of this accuracy predictor is only 0.37M, which is 3-4 orders of magnitude smaller than the inference MAC of the CNN classification models. The memory footprint of this accuracy predictor is only 5KB. Therefore, both the computation overhead and the memory overhead of the accuracy predictor are negligible.

# C   Cost Details

The on-device training cost of TinyTL and *FT-Full* on Pets is summarized in Figure 2 (left side), while the memory cost breakdown of TinyTL is provided in Figure 2 (right side). Compared to fine-tuning the full MobileNetV2-1.4, TinyTL not only greatly reduces the activation size but also reduces the parameter size (by applying weight quantization on the frozen parameters) and the training cost (by not updating weights of the feature extractor and using fewer training steps). Specifically, TinyTL reduces the memory footprint by $9.9\times$, and reduces the training computation by $27\times$ without loss of accuracy. Therefore, TinyTL is not only more memory-efficient but also more computation-efficient.

## Footnotes

[1]The training MAC of a sampled sub-network is roughly $2\times$ larger than its inference MAC, rather than $3\times$, since we do not need to update the weights of the main branches.

[2]Details of the accuracy predictor is provided in Appendix B.