[Reviews · NeurIPS 2020]

Review 1

Summary and Contributions: This paper proposed a memory-efficient method of transfer learning by freezing weights of pre-trained models and only updating bias. To maintain the ability to adapt and to choose a better feature extractor, this paper also proposed LiteResidual, a residual block that generates a residual feature map, and a feature extractor adapter.

Strengths: 1. The idea of freezing weights and partial updates on transfer learning is new and brings new insight for the transfer learning field with only updating the bias. 2. The demonstration of the method is well-organized.

Weaknesses: 1. Lack of Novelty: the novelty of the overall framework is not enough. It’s more like marginal contributions over the previous work Once-for-All (citation [1]) or common one-shot NAS method. The author didn’t well distinguish the proposed framework from directly applying Once-for-All on transfer learning with only updating bias. 2. Scalability of the claim of computation efficiency: The author claims the computation efficiency in the Section B of the supplement. To collect the training data for the accuracy predator, 450 subnets are trained on the 20% training dataset for only 1 epoch. That may get good performance on some easy datasets like the Flowers. But when facing more complex datasets requiring more training epochs, the computation cost for this part will increase a lot. And the author doesn’t give any explanation on how to face such scalability. 3. Not enough datasets are included for benchmark: Previous papers in transfer learning [7,27] not only benchmark in Cars, Flowers, and Aircraft, but also in more complex datasets like CIFAR-10 and Food-101 with more images. Compared with the previous work, the datasets included in this paper are not enough.

Correctness: Yes

Clarity: Yes

Relation to Prior Work: Yes

Reproducibility: Yes

Additional Feedback: More details in the Weaknesses. =================================== After rebuttal: Thanks for the clarification of all the concerned I had. And the added experiments on Food101 makes it a more solid work. And I am also glad to see the added ablation study on freezing biaes which make me think its a general and effiective techniques, and a key point in the proposed framework. And I agree R3 that recommending the authors to weave down the writing. I recommend acceptance.


Review 2

Summary and Contributions: The paper proposes tiny transfer learning, a method for adapting pre-trained models for new data on edge devices. The proposed method achieves this by not changing the model weights completely, but instead retraining only biases and augmenting the model with lite, residual 'corrections' to the feature map. The paper also proposes a featurizer selection method, where sub-networks form the pretrained super-net are identified and deployed depending on different datasets. The paper reports significant improvements in performance using their methods.

Strengths: Authors identify an important problem in edge deployment; adapting pre-trained models on device. Their main ideas are novel and lead to much improved memory utilization on the edge without sacrificing accuracy.

Weaknesses: Overall the paper introduces novel ideas to solve an important and interesting problem. The results are also impressive. Unfortunately, there are some weakness in terms of writing/details in the paper. The major weakness of this paper I found was a lack of clarity on the featurizer adaptation process. A lot of choices are unexplained, unmotivated or missing. Unfortunately, this takes away a lot from the rest of the paper as I am unable to get a sense of the complexity of this step. How expensive is this? How does one decide on various choices --- modelling, subnet selection, optimization steps, 'accuracy predictor', etc. For instance, + L53-54: What is discrete optimization space here? What are the variables? What is the objective? + L171-175: The notation is hard to follow. What are the elements of the set? How are they related? Though the intuition of sub-nets and supernets are clear, their definitions are not provided leading to a lot of questions. How does one decide what to choose as super network? Is it just the featurizer of our original network? How does one decide which subnets to even consider as candidates? Again 'discrete optimization space' is used her without defining what we are optimizing over. + L186-189: I'm not sure exactly what is happening in fine-tuning super-net set. What is the reason for randomly-sampling subnet in each training step? why is that superior over say sampling subnets based on their accuracy? (i.e. update better models more frequently) + L190: Not sure what "450, [sub-net, accuracy]" are. What is 'accuracy-predictor' ? This information is not provided in the main text and is unmotivated. Even the description in the appendix is only about the model structure. The 'why one requires it', 'how one decides on an architecture' etc are not discussed.

Correctness: The claims seem reasonable and the methodology seems correct.

Clarity: The paper could use another pass in terms of motivation and details. As mentioned previously, some details are wanting.

Relation to Prior Work: Yes

Reproducibility: Yes

Additional Feedback: On top of the comments/question raised previously, I only have two more: - Is adapting the biases necessary? Will just the residual models + subnet selection help? If it is unnecessary, will the method extend to non-linear smooth activation (sigmoid, tanh) ? - Related Work: In the efficient inference section of related work, it might be worth considering the papers that are part of the EdgeML project (https://github.com/microsoft/EdgeML). These are specific to edge devices and efficient inference. =========================================================== After rebuttal: Thank you for the wealth of details included in the rebuttal. Almost all of my major concerns have been addressed and I have modified by final score to reflect this. I would suggest including the main details to the final manuscript and taking another pass over writing just to improve flow. I recommend acceptance.


Review 3

Summary and Contributions: The paper presents a new Transfer learning pipeline/method which includes quite a few nice ideas either new or existing to enable on-device learning on memory-constrained edge devices where RAM costs a lot of energy. The paper makes the observation that during backprop bias updates require little memory (don't need activations to be stored) compared to weight updates. Combining this with new lite-residual blocks which are again cheap to update improves expressivity resulting in a better final model for downstream transfer tasks like Aircraft. Flowers and Cars. The authors also propose the use of one super-net to cater to all the downstream tasks and the appropriate subnet can be chosen through feature extractor adaptation routines that borrow ideas from the NAS literature. The experiments are in standard vision transfer learning settings and show that TinyTL has a much lesser memory footprint for transferring a pretrained network to a smaller downstream dataset with good boosts in accuracy.

Strengths: The paper tackles a very important problem of memory-efficient on-device training and the building blocks are well motivated, technically sound, and solid. Strengths include 1) Observations to update only biases 2) Adding new lite-residual blocks to gain the lost expressivity from not updating the weights 3) One super-net for pertaining and then adapting backbones according to the downstream datasets with feature extractor adaption. 4) Extensive experimentation and ablation studies along with memory and compute costs for all the experiments going on in the paper. 5) Details for reproducibility as well with a promise of open-sourcing the code. The only other recent paper I saw which tries to reduce memory footprint (through removed the expensive intermediate feature maps) is RNNPool (Saha et al., 2020) and I am not aware of other methods and will defer to other reviewers in case there is any. The 13.3X memory gains over Inception-V3 (Full fine-tuning) is very impressive. This paper also shows the trade-off between various modes of transfer learning like last layer tuning, BN+last layer and full and can be used for future benchmarks. They also show 2.3x and 9.8x memory reduction compared to the standard last layer and BN+last tuning methods while having better accuracy. The comparison to dynamic activation pruning is nice and the ablation about the design choices is encouraging. The paper also shows that the activation size reduces by 10x (again the only place with similar numbers against MBV2 is RNNPool) along with reduction in parameter size All the figures and tables are well made and the authors should be appreciated for that.

Weaknesses: These are not weaknesses but rather something I noticed and don't have clarity about. 1) I didn't understand the (ours) network in Figure 3. I looked around but didn't find what that was referring to. I assumed it referred to the TinyTL FeatureAdapt (FA) model from Table 1. 2) I again assume the Figure 4 is a theoretical computation and not an actual deployment on the RPi-1. It would be great to see an actual deployment if it is not already one. 3) The TinyTL method still assumes batch training, but on the edge devices smaller than RPi, things happen in a streaming fashion and probably batch size of 1 is what we might want to focus on. Any thoughts on this would be great. I think showcasing effectiveness in streaming would be a great thing assuming downstream dataset comes that way (which might be true in a lot of its devices). 4) There is some recent work on subnets inside a big net (like Worstamn et al 2020, which very recent and I don't expect the authors to know it) and training only BN layer and how effective they could be (https://arxiv.org/abs/2003.00152). It would be good to include them in related work broadly along with RNNPool kind of works. 5) I had to search around for the overhead due to the more involved pipeline with super-net and FA which is in Appendix, it would be great to point to it in the main paper and briefly mention it. 6) I don't completely get the parameter count in Fig 6 (right) can you flesh it out somewhere and it would be great (I don't know what model to use to compute and get that number). 7) Lastly, there are non-deep learning methods for on-device ML like Bonsai (Kumar et al., ICML 2017) and would be good to talk about them too. Authors should talk about negative impacts as well in the broader impact section.

Correctness: The method and claims are correct and empirical methodology is sound as well.

Clarity: I very much enjoyed reading the paper but it can benefit one more revision for grammar. The paper was well written, well-motivated, and easy to follow.

Relation to Prior Work: The paper does a good job of covering previous works and it is possible I might not know all of them and will defer to other reviewers if they find any issues on that front.

Reproducibility: Yes

Additional Feedback: I am very much willing to change the score (so the initial score is not the final score) based on rebuttal on the concerns I have and reviewer discussion. =========================================================== After rebuttal: Thanks for the clarifications on streaming training, hardware deployment and the confusion in references to figures and parameter counts. The experiments with GN for batch 1 seem promising. I agree with R1 about the missing couple of datasets, I appreciate the Table A in the rebuttal for Food-101 and comparison to biases. The paper contains a lot of information and is hard to process. This has led to reviewers miss key points. I recommend authors to weave down the writing. The appendix of the paper has a lot of the details and linking them appropriately might help in the future. I recommend acceptance.


Review 4

Summary and Contributions: This manuscript introduces Tiny-Transfer-Learning to address memory-constrained issues on edge devices. The proposed method adapts pre-trained models to newly collected data by freezing the weights, but not biases. Moreover, it suggests augmenting a lite residual module and selecting an architecture of feature extractor from a largely pre-trained super-net. The experimental results outperform fine-tuning methods significantly.

Strengths: This manuscript tackles an important problem of training on edge devices; back-propagation causes a huge training memory footprint. The proposed method is novel and leads a lot of improvement in reducing memory, instead of parameters. I expect the proposed feature extractor adaptation to be applicable to conventional transfer learning.

Weaknesses: Methodology (feature extractor adaptation) - The proposed method seems to rely heavily on feature extractor adaptation to avoid scarifying accuracy. But, the expense of this process is unclear. Is fine-tuning the super-net is available on edge devices? If not, it conflicts with the addressing problem of memory-constrained on-device learning. Ablations - The proposed method is configured by three components; updating bias, lite residual learning, feature extractor adaptation. To understand the proposed method in detail, component-wise ablation study is needed to resolve the following questions; 1) Is the updating bias important? What happens when the bias is frozen? 2) Could the proposed method avoid scarifying accuracy without the feature extractor adaptation?

Correctness: I doubt a part of the methodology (fine-tuning super-net stage in feature extractor adaptation process) violence the tackled problem as mentioned above. The rest seems to correct.

Clarity: There is a considerable lack of clarity on feature extractor adaptation.

Relation to Prior Work: There is no proper previous contribution to discuss because it seems to the first work on the tackled problem.

Reproducibility: Yes

Additional Feedback: I have one more question, - Could baseline methods (fine-tuning) share the benefit of the feature extractor adaptation? I expect it could be broadly applicable to conventional transfer learning. =========================================================== After rebuttal: Thanks for the clarification for the questions I had, `correctness section` (cost of fine-tuning super-net), and more ablation study. In applying feature extractor adaptation (FA) for transfer learning, it seems great that conventional transfer learning also shares the benefits of FA, while the `FA+Full' in rebuttal outperforms the reported value of `FA+TinyTL' in the manuscript. I would suggest including the `FA+transfer learning' to the final manuscript and clarifying the contribution of each components of this work;updating bias, lite residual learning, feature extractor adaptation. I recommend acceptance.

[Author Response · NeurIPS 2020]

We thank all reviewers for their comments and thank R2 and R3 for suggesting the literature. We will revise our paper accordingly. The presentation will be polished and more discussions will be included in the broader impact section.

**R1:** Novelty compared to Once-for-All. This paper targets at solving an **entirely new** challenge: on-device transfer learning on memory-constrained edge devices, which is **fundamentally different** from existing NAS for efficient inference problem in Once-for-All. Whether and how can Once-for-All help towards addressing this new problem is an open research question and **never** explored before. Only updating biases along with the memory-saving insights behind it (Sec. 3.1) are also **entirely new**. To our best knowledge, we are the **first** to introduce this finding. Moreover, based on the insights from only updating biases, we further designed a new technique 'Lite Residual Learning', which **efficiently** recovers the lost expressivity from not updating the weights. The effectiveness of our method has been **thoroughly verified** (**9.5-12.5**× memory saving on multiple datasets in Fig.4, up to **13.3**× memory saving in Tab.1). We believe our findings will open up new opportunities for on-device learning.

**R1:** Scalability of the computation efficiency and results on more datasets. Our approach might be misunderstood by the reviewer. First, we **never** train sub-nets when collecting the training data for the accuracy predictor; **instead**, we directly inherit the weights from the super-net to initialize the sub-nets, thus scalable to large datasets. Second, this work targets at on-device transfer learning (much less data/memory), **not** conventional transfer learning. Therefore, we focus on datasets with fewer images (e.g., Flowers) that are **much closer** to real-world on-device scenarios than large datasets. Certainly, our method **generalizes** to large datasets. In Table A, we justify the effectiveness of TinyTL on Food101 (the largest dataset in [7, 27]). TinyTL **consistently** achieves **significant** memory saving (**7.3**×) with little accuracy loss.

|  | Mem. | Food101 |
|---|---|---|
| Full [27] | 802MB | 87.7% |
| TinyTL | 109MB | 87.2% |

Table A: Results on Food101.

**R2:** Details of feature extractor adaptation. We will add more details to the main paper in the final version. **(L53-54)** The discrete optimization space includes depth ('Repeat': 1,2,3), width ('Expand Ratio': 3,4,6) and kernel size ('Kernel Size': 3,5,7) [Appendix E]. Each architecture configuration corresponds to a sub-net. The objective is to find the best sub-net that maximizes transfer accuracy. **(L171-175)** The super-net is a normal neural network with the maximum depth, width, and kernel size. Sub-nets are derived from the super-net by sparsely activating parts of the model according to the architecture configuration. Specifically, consider a 7x7 conv layer denoted as $W_{0:c1,0:c2,0:7,0:7}$, an example of the candidate weight operation set (in Eq.5) is $\{W_{0:c1,0:c2,0:7,0:7}, W_{0:c1,0:c2,1:6,1:6}, W_{0:c1,0:c2,2:5,2:5}\}$, which corresponds to kernel size = 7/5/3. **(L186-189)** In the process of fine-tuning the super-net, we only **update the memory-efficient modules** (bias, lite residual, classifier head), while **freezing the memory-heavy modules**. Since sub-nets inherit weights from the super-net, all sub-nets are adapted to the target dataset while keeping the memory footprint small. **Random sampling** can ensure each sub-net is evenly trained, while accuracy-based sampling biases towards *early* good performers and keeps sampling them more frequently without exploring others. A sub-net that performs well early does not guarantee to be the best in the end. Therefore we chose random sampling. **(L190)** The accuracy predictor can predict the transfer accuracy given a sub-net architecture. Conventionally, we need to evaluate many sub-nets on the target dataset to find the best one, which is expensive. Instead, we exploit a highly efficient accuracy predictor [Appendix C] to reduce the cost. '450 [sub-net, accuracy]' is the collected dataset for training the accuracy predictor.

**R2, R5:** Cost of feature extractor adaptation. We have **strong** reasons to believe that the whole feature extractor adaptation process (including fine-tuning the super-net) is feasible on edge devices [Appendix B]. First, as we freeze the weights of the feature extractor, the peak memory cost of fine-tuning the super-net is **only** 64MB under batch size 8, which is 4x smaller than the DRAM size of RPi-1. Moreover, combined with group normalization (refer to '**R3:** Streaming Training'), TinyTL can support training with batch size 1, where the peak memory cost is **only** 26MB. It allows fitting the whole process into the on-chip SRAM of TPU, which is **128x** energy-efficient than DRAM (Fig.1). Second, our total computational cost is **18x** smaller than fine-tuning the full network [27] while preserving accuracy.

**R2, R5:** Effects of freezing biases. Adapting biases is necessary. Without it, the accuracy drops by 1.7% on Cars, 0.5% on Flowers, and 4.1% on Aircraft (Table B).

|  | Cars | Flowers | Aircraft |
|---|---|---|---|
| w/ bias | 91.6% | 97.5% | 84.0% |
| w/o bias | 89.9% | 97.0% | 79.9% |

Table B: Effects of freezing biases.

**R5:** Results without feature extractor adaptation. If disabling the feature extractor adaptation, the accuracy drops by 2.2% on Cars, 0.6% on Flowers, and 2.5% on Aircraft (shown in Tab.1, page7). Feature extractor adaptation is critical.

**R5:** Apply to conventional transfer learning. 'Fine-tuning the full network' can also benefit from feature extractor adaptation (FA). Compared to InceptionV3+Full, FA+Full improves the accuracy from 91.3% to 93.2% on Cars, from 96.3% to 98.3% on Flowers, from 85.5% to 88.9% on Aircraft. We will include this feature in code release.

**R3:** Streaming Training. TinyTL supports streaming training by replacing batch normalization (BN) with group normalization (GN), which supports batch 1 training. We observe little loss of accuracy from BN to GN: 89.4%->89.0% on Cars, 96.9%->96.7% on Flowers, 81.5%->81.1% on Aircraft. We will include the new results in the revision.

**R3:** Hardware deployment. Fig.4 used theoretical values as Pytorch does not support fine-grained memory management. We target co-designing the on-device training framework to fully exploit the theoretical benefits, which is beyond the scope of this paper. We will make this clear in the revision.

**R3:** Fig 3, Fig 6. In Fig.3, 'ours' refers to TinyTL FA from Tab.1. In Fig.6, the parameter size consists of two parts: i) frozen parameters (2.3MB,8bits); ii) trained parameters (11.3MB,32bits). We will make it more clear in the revision.

[Meta-Review · NeurIPS 2020]

All four referees support accept. At some angle, AC thinks that novelty might be a bit limited and the proposed scheme is of engineering type. However, as pointed out by reviewers, AC also agree that the problem studied in this paper is important/interesting, and the ideas/results are practical/impressive. Hence, AC recommends acceptance. AC also suggests the authors to clarify the write-ups following the reviewers' suggestions in the final draft.